# Coupled Simulation of a Vacuum Creation System and a Rectification Column Block

**Eduard Osipov ***[ID]**, Eduard Telyakov and Sergey Ponikarov**

Mechanical Engineering for Chemical Industry, Mechanical Engineering, Kazan National Research Technological University, 420015 Kazan, Russia; Tesh1939@mail.ru (E.T.); mahp_kstu@mail.ru (S.P.)
* Correspondence: eduardvosipov@gmail.com; Tel.: +7-(917)-293-66-00

**Abstract:** The purpose of this study was the coupling simulation of the vacuum block of the ethanolamine mixture separation unit to determine the optimal layout of the vacuum creation system. For this, a computational model of the vacuum unit, which was identified by comparing the computational data with the data of an industrial study of vacuum rectification columns, was synthesized in the Unisim Design R461 software package. To determine the required load on the vacuum system, a numerical experiment was carried out, during which it was discovered that the load on the system would be 9600 m$^3$/h. It was proposed to replace individual column vacuum pumps with a single vacuum-generating system (VGS) based on a liquid ring vacuum pump (LRVP). When defining the layout, two possible schemes were considered, the models of which were created in Unisim Design R461. The system layout was determined by matching the characteristics of the system elements with the characteristics of the vacuum columns. A technical and economic comparison of the proposed solutions was carried out and the payback period for capital costs was calculated, which for Scheme 1 was 4.14 years, and for Scheme 2–3.59 years.

**Keywords:** vacuum creating system; steam ejection pumps; liquid ring vacuum pump; universal modeling program; rectification

## 1. Introduction

At chemical plants, ethanolamines are produced by oxyethylation of ammonia using water as a catalyst [1–3]. The product stream produced from the reactor block consists of MEA (monoethanolamine), DEA (diethanolamine), and TEA (triethanolamine), which are separated into components in the rectification columns [4–6]. Due to the low thermal stability of the separated components, as well as rather strict requirements for the resulting products, the rectification process is carried out under vacuum, with a residual pressure of up to 6.65 mbar [7]. The steam ejection pumps (SEP) used as vacuum-generating systems (VGSs) are obsolete and are characterized by increased consumption of expensive energy resources [8], so it was decided to replace the existing vacuum pumps with a single VGS based on the liquid ring vacuum pump (LRVP) [9]. The system layout was determined by mathematical modeling using UMP (universal modeling program) tools.

Analysis of complex objects of chemical industries is usually carried out using the principle of dismemberment (decomposition) of a CCTS (complex chemical and technological system) into its constituent elements (blocks) of a lower level of hierarchy (input) [10–14]. Blocks are characterized by a fairly high level of autonomy, which allows us to operate in terms of their characteristics [13], which determine the main properties of blocks. Direct modeling is performed in the environments of UMP—ChemCad, Aspen, Unisim Design, etc. [10,11]. The connection between blocks is usually determined by the input, output, and circulating inter-block flows that occur in the CCTS. The impact

of these relationships is determined by mathematical means of the UMP at the level of calculating information about expenditures and parameters of the state of these flows.

However, in the process of modeling elements of individual blocks, there is a need to fix individual parameters of mathematical models of elements. The results of modeling will depend on the correct assignment of these parameters, which are difficult to assess at the stage of forming a mathematical model of an individual element. In [12], it is proposed to represent the dependence between input, output, and internal parameters of the model as generalized characteristics that determine the main properties of the block. In this case, the task of simulating CCTS is simplified, but the dependence of the result on the reliability of calculating the interface conditions for individual blocks is even more complicated.

The study of the conditions for the coupling of individual elements of a complex chemical technological object with each other, the effect of these conditions on the properties of the object, as well as the comparison of various ways to reduce the load on the forevacuum pump was the purpose of this work.

## 2. Structural Analysis of the Production in Question

In the production under consideration, the separation of the amine mixture into target products is carried out in three rectification columns—K-40, K-56, and K-92, equipped with four-stage SEP, the working body of which is medium-pressure water vapor. The structural diagram of the production in question is shown in Figure 1a.

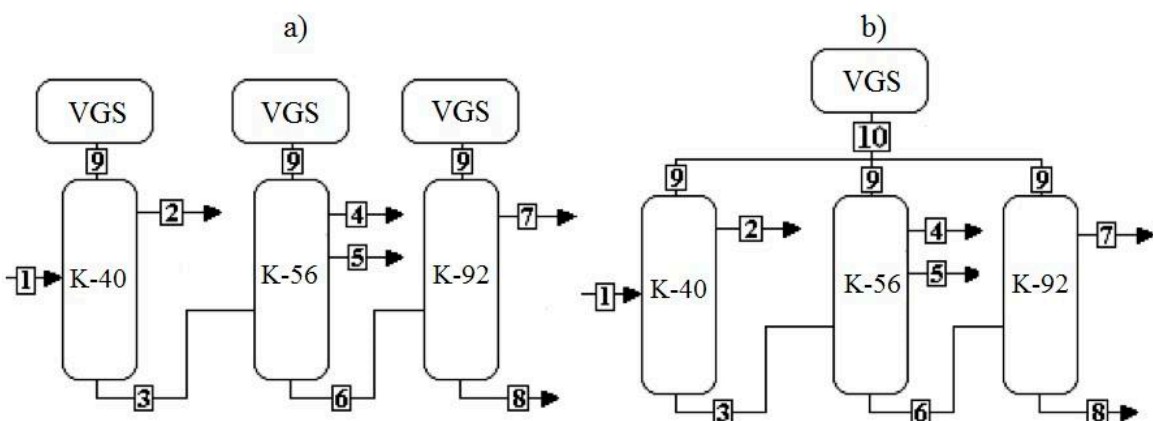

**Figure 1.** Block diagrams of the existing (**a**) and proposed (**b**) blocks. Flows: 1—bottom product of column 29; 2—flow of MEA; 3—DEA of grade B; 4—MEA+water; 5—DEA of grade A; 6—TEA; 7—TEA of grade A; 8—residue; 9—gas flow to the vacuum-generating systems (VGS) from the columns; 10—total flow to the VGS.

The pipe space of the intermediate condensers is fed with "stagnant" recycled water, and condensate and non-condensed gases are removed from the inter-tube space. Thus, the following energy resources are consumed during the operation of the SEP [14–16]: water vapor and recycled water. Since the ejector is shifting the working fluid and the pumped medium, which contains ethanolamines, the condensate coming out of the SEP can be considered as chemically contaminated.

Based on all of the above, we can conclude that in relation to the production under consideration, the SEP is outdated and unecological, so its replacement with a modern energy-efficient and environmentally friendly VGS is an urgent task. When evaluating the feasibility study of the proposed solution to improve the existing VGS, it is necessary to take into account the costs of the enterprise for the production of steam and recycled water, as well as for the disposal of CCS (chemically contaminated substance). Prices for these resources are shown in Table 1 (provided by consumer).

**Table 1.** Resources prices.

| Resource | Value | Unit of Measurement |
|----------|-------|---------------------|
| Water vapor | 17.89 | $/Gcal |
| Circulating water | 0.02 | $/m$^3$ |
| Electricity | 0.04 | $/kW |
| Clearing CCS | 0.22 | $/m$^3$ |

Based on the experience of such works as [15–20], the most optimal solution for the reconstruction of existing VGS is to replace the SEP with a single VGS column based on the LRVP. The proposed block diagram is shown in Figure 1b.

To determine the layout of the proposed VGS, it is necessary to calculate the flow rate and composition of the mixture entering the VGS, that is, it is necessary to determine the load on the VGS. This load will be determined by the condensation conditions in the inter-tube space of the "tail condensers", which depend on the operating conditions of the installation. The parameters that determine the efficiency of condensation of the medium are the temperature and pressure of condensation in the inter-tube space, which are not directly measured and controlled at the production site. Therefore, these parameters are selected based on the results of the installation survey, and the calculation itself is carried out with the wide use of mathematical modeling tools [21,22] in the UMP environment. The software product Unisim Design R461 was chosen as the UMP, which allows us to solve such problems fairly accurately [7].

The calculation results are significantly influenced by the adopted package for calculating the vapor–liquid equilibrium. The program database includes the Amine package, which is based on the Kent–Eisenberg model. The Kent–Eisenberg model is a simplified way to model the reactions (and phase equilibrium) in a gas sweetening system. The model is used in a system where water with one amine is used to treat gas with carbon dioxide, sulfuric acid, and/or ammonia [23].

In [24], an acid gas package was used, however, within the framework of the problem posed in the article; the application of this method is difficult, since the pumped mixture does not contain the components $CO_2$, $H_2S$, etc. Therefore, according to the recommendations [25–27], the UNIFAC (UNIQUAC Functional-group Activity Coefficients) model was chosen, the enthalpy model—Amine.

In the UNIFAC K model [25], the liquid phase activity coefficients for each species are calculated from the UNIFAC group contribution method. The limitations of UNIFAC are temperature range from 275 K to 425 K and pressure up to a few atmospheres. Moreover, this UNIFAC reads the group contribution parameter stored in the VLE (vapor-liquid Equilibrium) database. Table 2 presents UNIFAC Group Interaction Parameters (calculated in Unisim Design R461).

The UNIFAC model splits up the activity coefficient for each species in the system into two components: a combinatorial $\gamma^c$ and a residual component $\gamma^r$:

$$ln\gamma_i = ln\gamma_i^c + ln\gamma_i^r \tag{1}$$

$$ln\gamma_i^c = ln\frac{\varphi_i}{x_i} + \frac{z}{2}q_i ln\frac{\theta_i}{\varphi_i} + L_i - \frac{\varphi_i}{x_i}\sum_{j=1}^{n} x_j L_j \tag{2}$$

$$\theta_i = \frac{x_i q_i}{\sum_{j=1}^{n} x_j q_j} \tag{3}$$

$$\tau_i = \frac{x_i r_i}{\sum_{j=1}^{n} x_j r_j} \tag{4}$$

$$L_i = \frac{z}{2}(r_i - q_1) - (r_i - 1) \tag{5}$$

$$r_i = \sum_{k=1}^{n} v_k R_k \tag{6}$$

$$q_i = \sum_{k=1}^{n} v_k Q_k \tag{7}$$

$$ln\gamma_i^r = \sum_{k}^{n} v_k^{(i)}\left[\boldsymbol{ln}\Gamma_k - ln\Gamma_k^{(i)}\right] \tag{8}$$

$$ln\Gamma_k = Q_k\left[1 - \ln\sum_m \Theta_m\Psi_{mk} - \sum_m \frac{\Theta_m\Psi_{mk}}{\sum_n \Theta_n\Psi_{nk}}\right] \tag{9}$$

$$\Theta_m = \frac{Q_m X_m}{\sum_n Q_n X_n} \tag{10}$$

$$\Psi_{mk} = \exp\left[-\frac{U_{mn} - U_{nm}}{RT}\right] \tag{11}$$

$$X_m = \frac{\sum_j v_m^j x_j}{\sum_j \sum_n v_n^j x_j} \tag{12}$$

$$\Psi_{mn} = \exp\frac{-a_{mn}}{T} \tag{13}$$

VLE data calculated by Unisim Design R461 for pressure 4 mbar is shown in Tables 3–5.

**Table 2.** UNIF Group Interaction Parameters.

| Grp$_i$ | Grp$_j$ | A$_{ij}$ | A$_{ji}$ | B$_{ij}$ | B$_{ji}$ | C$_{ij}$ | C$_{ji}$ |
|---|---|---|---|---|---|---|---|
| $H_2O$ | $CH_2$ | 300 | 1318 | 0 | 0 | 0 | 0 |
| $H_2O$ | OH | −229.1 | 353.5 | 0 | 0 | 0 | 0 |
| $H_2O$ | $CNH_2$ | 48.89 | −330.4 | 0 | 0 | 0 | 0 |
| $H_2O$ | CNH | 168 | −448.2 | 0 | 0 | 0 | 0 |
| $H_2O$ | $(C)_3N$ | 304 | −598.8 | 0 | 0 | 0 | 0 |
| $CH_2$ | OH | 985 | 156.4 | 0 | 0 | 0 | 0 |
| $CH_2$ | $CNH_2$ | 391 | −30.48 | 0 | 0 | 0 | 0 |
| $CH_2$ | CNH | 255.7 | 65.33 | 0 | 0 | 0 | 0 |
| $CH_2$ | $(C)_3N$ | 206.6 | −83.98 | 0 | 0 | 0 | 0 |
| OH | $CNH_2$ | 8.64 | −242.8 | 0 | 0 | 0 | 0 |
| OH | $(C)_3N$ | 42.7 | −150 | 0 | 0 | 0 | 0 |
| OH | $(C)_3N$ | −323 | 28.6 | 0 | 0 | 0 | 0 |
| $CNH_2$ | CNH | −107.2 | 127.4 | 0 | 0 | 0 | 0 |
| $CNH_2$ | $(C)_3N$ | −41.11 | 48.89 | 0 | 0 | 0 | 0 |
| CNH | $(C)_3N$ | −189.2 | 865.9 | 0 | 0 | 0 | 0 |

**Table 3.** XY data for Water/MEA.

| t (°C) | $x_1$ | $y_1$ | $\gamma_1$ | $\gamma_2$ | $\varphi_1$ | $\varphi_2$ |
|--------|-------|-------|------------|------------|-------------|-------------|
| 58.399 | 0 | 0 | 0.66 | 1 | 1 | 1 |
| 46.768 | 0.05 | 0.57312 | 0.655 | 0.999 | 1 | 1 |
| 39.064 | 0.1 | 0.77172 | 0.659 | 0.996 | 1 | 1 |
| 33.41 | 0.15 | 0.86181 | 0.669 | 0.991 | 1 | 1 |
| 28.956 | 0.2 | 0.90998 | 0.683 | 0.983 | 1 | 1 |
| 25.277 | 0.25 | 0.93861 | 0.699 | 0.973 | 1 | 1 |
| 22.132 | 0.3 | 0.95688 | 0.718 | 0.958 | 1 | 1 |
| 19.379 | 0.35 | 0.96914 | 0.738 | 0.941 | 1 | 1 |
| 16.924 | 0.4 | 0.97767 | 0.76 | 0.919 | 1 | 1 |
| 14.707 | 0.45 | 0.98375 | 0.784 | 0.893 | 1 | 1 |
| 12.683 | 0.5 | 0.98818 | 0.808 | 0.863 | 1 | 1 |
| 10.823 | 0.55 | 0.99144 | 0.834 | 0.828 | 1 | 1 |
| 9.107 | 0.6 | 0.99385 | 0.859 | 0.789 | 1 | 1 |
| 7.519 | 0.65 | 0.99565 | 0.885 | 0.744 | 1 | 1 |
| 6.052 | 0.7 | 0.99698 | 0.91 | 0.696 | 1 | 1 |
| 4.7 | 0.75 | 0.99797 | 0.934 | 0.644 | 1 | 1 |
| 3.464 | 0.8 | 0.99869 | 0.956 | 0.589 | 1 | 1 |
| 2.346 | 0.85 | 0.99921 | 0.975 | 0.532 | 1 | 1 |
| 1.351 | 0.9 | 0.99957 | 0.989 | 0.476 | 1 | 1 |
| 0.486 | 0.95 | 0.99983 | 0.997 | 0.424 | 1 | 1 |
| −0.255 | 1 | 1 | 1 | 0.38 | 1 | 1 |

**Table 4.** XY data for MEA/DEA.

| t (°C) | $x_1$ | $y_1$ | $\gamma_1$ | $\gamma_2$ | $\varphi_1$ | $\varphi_2$ |
|--------|-------|-------|------------|------------|-------------|-------------|
| 136.648 | 0 | 0 | 0.73 | 1 | 1 | 1 |
| 113.855 | 0.05 | 0.75205 | 0.734 | 0.999 | 1 | 1 |
| 102.268 | 0.1 | 0.8903 | 0.746 | 0.997 | 1 | 1 |
| 94.776 | 0.15 | 0.93876 | 0.76 | 0.993 | 1 | 1 |
| 89.282 | 0.2 | 0.96156 | 0.777 | 0.988 | 1 | 1 |
| 84.951 | 0.25 | 0.97416 | 0.794 | 0.98 | 1 | 1 |
| 81.378 | 0.3 | 0.98187 | 0.812 | 0.971 | 1 | 1 |
| 78.338 | 0.35 | 0.98691 | 0.831 | 0.959 | 1 | 1 |
| 75.694 | 0.4 | 0.99037 | 0.85 | 0.945 | 1 | 1 |
| 73.358 | 0.45 | 0.99283 | 0.868 | 0.929 | 1 | 1 |
| 71.269 | 0.5 | 0.99463 | 0.886 | 0.91 | 1 | 1 |
| 69.384 | 0.55 | 0.99597 | 0.904 | 0.889 | 1 | 1 |
| 67.673 | 0.6 | 0.99699 | 0.921 | 0.866 | 1 | 1 |
| 66.112 | 0.65 | 0.99776 | 0.937 | 0.84 | 1 | 1 |
| 64.686 | 0.7 | 0.99836 | 0.952 | 0.812 | 1 | 1 |
| 63.38 | 0.75 | 0.99883 | 0.965 | 0.782 | 1 | 1 |
| 62.185 | 0.8 | 0.99919 | 0.977 | 0.749 | 1 | 1 |
| 61.093 | 0.85 | 0.99948 | 0.987 | 0.714 | 1 | 1 |
| 60.101 | 0.9 | 0.9997 | 0.994 | 0.678 | 1 | 1 |
| 59.203 | 0.95 | 0.99987 | 0.998 | 0.639 | 1 | 1 |
| 58.399 | 1 | 1 | 1 | 0.599 | 1 | 1 |

**Table 5.** XY data for DEA/TEA.

| T (°C) | $x_1$ | $y_1$ | $\gamma_1$ | $\gamma_2$ | $\varphi_1$ | $\varphi_2$ |
|---|---|---|---|---|---|---|
| 189.122 | 0 | 0 | 0.984 | 1 | 1 | 1 |
| 179.96 | 0.05 | 0.40237 | 0.985 | 1 | 1 | 1 |
| 173.28 | 0.1 | 0.60229 | 0.986 | 1 | 1 | 1 |
| 168.114 | 0.15 | 0.71684 | 0.988 | 1 | 1 | 1 |
| 163.944 | 0.2 | 0.78917 | 0.989 | 0.999 | 1 | 1 |
| 160.469 | 0.25 | 0.83814 | 0.99 | 0.999 | 1 | 1 |
| 157.503 | 0.3 | 0.87303 | 0.991 | 0.998 | 1 | 1 |
| 154.925 | 0.35 | 0.89891 | 0.992 | 0.998 | 1 | 1 |
| 152.649 | 0.4 | 0.91869 | 0.994 | 0.997 | 1 | 1 |
| 150.618 | 0.45 | 0.93421 | 0.995 | 0.996 | 1 | 1 |
| 148.785 | 0.5 | 0.94664 | 0.996 | 0.995 | 1 | 1 |
| 147.118 | 0.55 | 0.95676 | 0.996 | 0.994 | 1 | 1 |
| 145.592 | 0.6 | 0.96512 | 0.997 | 0.993 | 1 | 1 |
| 144.186 | 0.65 | 0.97212 | 0.998 | 0.992 | 1 | 1 |
| 142.884 | 0.7 | 0.97804 | 0.998 | 0.99 | 1 | 1 |
| 141.672 | 0.75 | 0.9831 | 0.999 | 0.989 | 1 | 1 |
| 140.54 | 0.8 | 0.98746 | 0.999 | 0.988 | 1 | 1 |
| 139.479 | 0.85 | 0.99124 | 1 | 0.986 | 1 | 1 |
| 138.48 | 0.9 | 0.99454 | 1 | 0.985 | 1 | 1 |
| 137.539 | 0.95 | 0.99744 | 1 | 0.983 | 1 | 1 |
| 136.648 | 1 | 1 | 1 | 0.982 | 1 | 1 |

## 3. Calculation Diagrams of the Main Elements of the Block

**A preliminary analysis of the data**. As noted above, the separation of the ethanolamine mixture into target components occurs in a chain of rectification columns operating under vacuum. The existing vacuum system is currently functioning stably; however, the pressure at the top of the vacuum columns differs from that originally laid down in the design. The system itself is morally and physically outdated today and needs to be replaced. The technological parameters of the plants are shown in Tables 6–8.

**Table 6.** Technological parameters and calculation data for the K-40 column.

| Stream | Composition, Mass Fraction | | | | Temperature, °C | Mass Rate, kg/h |
|---|---|---|---|---|---|---|
| | **Water** | **MEA** | **DEA** | **TEA** | | |
| Distillate (o) | 0.001–0.2 | 0.98–0.999 | 0.009–0.07 | - | 30–40 | 900–1340 |
| Distillate (s) | 0.001 | 0.99 | 0.009 | - | 35 | 1247 |
| Reflux (o) | 0.001–0.2 | 0.98–0.999 | 0.009–0.07 | - | 30–40 | 100–300 |
| Reflux (s) | 0.001 | 0.99 | 0.009 | - | 35 | 260 |
| Feed (o) | 0.0004 | 0.502 | 0.317 | 0.1806 | 60–100 | 1900–2700 |
| Feed (s) | 0.00037 | 0.501 | 0.313 | 0.181 | 90 | 2554 |
| Bottom (o) | 0.0001–0.003 | 0.015–0.037 | 0.55–0.72 | 0.39–0.48 | 141–150 | 1000–1360 |
| Bottom (s) | - | 0.0158 | 0.621 | 0.361 | 147 | 1285 |

**Table 7.** Technological parameters and calculation data for the K-56 column.

| Stream | Composition, Mass Fraction | | | | Temperature, °C | Mass Rate, kg/h |
|---|---|---|---|---|---|---|
| | Water | MEA | DEA | TEA | | |
| Distillate (o) | 0.001–0.002 | 0.29–0.79 | 0.29–0.79 | ≤0.0005 | 70–90 | 550–750 |
| Distillate (s) | - | 0.11 | 0.89 | - | 90 | 323 |
| Reflux (o) | 0.001–0.002 | 0.29–0.79 | 0.29–0.79 | ≤0.0005 | 60–80 | 350–650 |
| Reflux (s) | - | 0.11 | 0.89 | - | 80 | 367 |
| Side product (o) | 0.1 | 0.006–0.01 | 0.98–0.992 | 0.001–0.002 | 140–160 | 480–660 |
| Side product (s) | - | 0.0012 | 0.985 | 0.014 | 147 | 546 |
| Feed (o) | 0.0001–0.003 | 0.015–0.037 | 0.55–0.72 | 0.39–0.48 | 95–105 | 1000–1360 |
| Feed (s) | - | 0.0158 | 0.621 | 0.361 | 105 | 1285 |
| Bottom (o) | 0.001–0.002 | 0.001–0.002 | 0.065–0.09 | 0.90–0.945 | 195–205 | 320–440 |
| Bottom (s) | - | - | 0.065 | 0.935 | 206 | 490 |

**Table 8.** Technological parameters and calculation data for the K-92 column.

| Stream | Composition, %. | | | | Temperature, °C | Mass Rate, kg/h |
|---|---|---|---|---|---|---|
| | Water | MEA | DEA | TEA | | |
| Distillate (o) | 0.001 | 0.001 | 0.01–0.02 | 0.97–0.98 | 90–110 | 290–400 |
| Distillate (s) | 0.0011 | 0.001 | 0.0189 | 0.979 | 90 | 382 |
| Reflux (o) | 0.001 | 0.001 | 0.01–0.02 | 0.97–0.98 | 80–100 | 200–300 |
| Reflux (s) | 0.0011 | 0.001 | 0.0189 | 0.979 | 90 | 300 |
| Feed (o) | 0.001–0.002 | 0.001–0.002 | 0.065–0.09 | 0.9–0.945 | 195–205 | 322–440 |
| Feed (s) | - | - | 0.065 | 0.935 | 206 | 490 |

**Calculation scheme of the distillation columns**. After distilling water and residual ammonia in the K-29 column, the dehydrated flow of ethanolamines is fed between the second and third sections of the K-40 column nozzle and is divided into monoethanolamine (distillate) and diethanolamine grade B (bottom). The vapors that do not condense in the condenser enter the tail condenser, which is cooled by a stream of cooled refrigerant. Those gases that are not condensed in the "tail" condenser are pumped out by a four-stage SEP, which creates a vacuum in the rectification column.

Simulation of the rectification plant was carried out in the UMP Unisim Design R461, in which the design scheme of columns was synthesized (Figure 2). The main task of computer simulation of the process was to determine the composition and flow of gases entering the VGS. To calculate the column in the program database, there are various modules of rectification columns that differ from each other in their mathematical description. Based on the recommendations of the program, the distillation column module was used to calculate the column. For convergence of the calculation, the following data must be entered in the module specification: column pressure, pressure drop, number of theoretical plates, the feed plate number, operating modes of the condenser, and cube. The correctness of the specified parameters was determined by comparing the calculated data with the data of the industrial inspection of columns (provided by consumer).

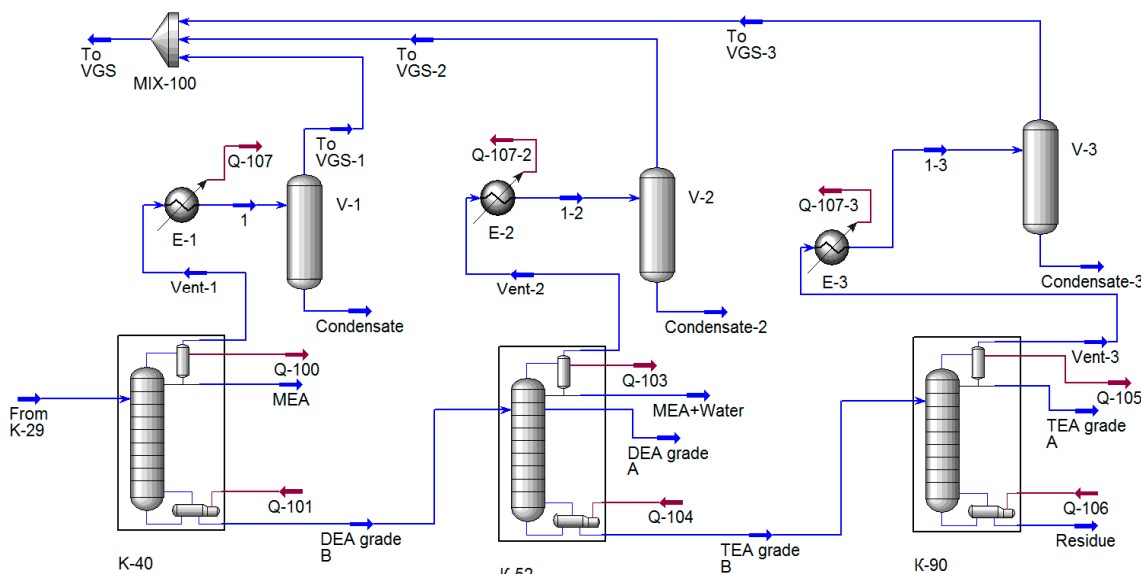

**Figure 2.** Calculation diagram of the amine mixture separation unit synthesized in the UMP Unisim Design R461.

The tail capacitors were set by the separator and cooler modules, which set the final temperatures of the heat carriers. It was this module that formed the flow going to the VGS.

Next, the cubic residue of the K-40 column enters the K-52 column, which is designed to produce A-grade diethanolamine, which is removed from the installation by a side stripping. Water vapor, MEA, and DEA from the top of the column condense in a condenser, are withdrawn to the collector, from which, part of the condensate is returned in the form of phlegm irrigation. The vapors that do not condense in the condenser enter the "tail" condenser, which forms the final load on the VGS. The heat required for the rectification process is supplied to a film evaporator heated by steam.

The simulation of the rectification plant was carried out using the same modules as the K-40 column simulation, except that the side selection was added to the module 2 specification, which was set by the selection plate and mass flow rate. Numeric values were set in such a way as to ensure that they correspond to production values.

The K-92 column is designed for the production of Grade A triethanolamine with improved performance in appearance from the cubic liquid of the K-56 column. Power is supplied to the distribution plate located between the nozzle layers. The supply of heat is carried out using a film evaporator, vapor from columns is received in the condenser, condense, and drop on to the dead plate. The resulting liquid is partially returned to the column as reflux, and the distillate is removed from the unit. A comparison of calculated parameters with production data is presented in Tables 6–8.

**Calculation scheme of the VGS**. The existing VGS are four-stage steam-ejector VGS, which are installed on each column. Each SEP is designed to pump 10 kg/h of the mixture (9.5 kg/h of air and 0.5 kg/h of amines) at a suction pressure of 1.33 mbar. However, according to production data, at the moment the pressure in the vacuum columns is 10–15 mbar, which indicates that the VGS is currently being operated in a zone of significant overload. The most likely reason for this is the discrepancy between the production conditions and the characteristics of the VGS.

The value of the pressure on the suction in the VGS, measured by the standard gauge, is $-1$ kg/cm$^2$ (overpressure), which indicates the "scale" of the device. Therefore, the VGS layout based on the characteristics of the existing SEP can lead to significant errors.

Designing a vacuum system for a technological object is a laborious task with a high proportion of difficult to determine parameters (flow rate and composition of the pumped mixture, the size of the vacuum pump components, characteristics of the object, etc.), the exact calculation of which is possible

only with the extensive use of computer technology, in including special programs for modeling chemical–technological systems.

When carrying out work of this kind, given the complexity of the systems under consideration, engineers are forced to rely not on mathematical modeling, but on some expert assessments and judgments when choosing one scheme or the other. In addition, there are difficulties in assessing the functioning of vacuum systems of various kinds since these systems consist of various equipment.

To date, no unified design methodology has been formulated for vacuum systems for process plants. There is also no single methodology for assessing the effectiveness of various types of VGS. In [28], recommendations are given on the field of application of various types of vacuum pumps in the area of developed vacuum; however, they do not contain clear recommendations for the design of these systems.

As a rule, the designers of vacuum systems and operating personnel for the VGS type rely on some production experience, while often the design does not take into account the features of the evacuated installation itself, which often leads to a failure to achieve the set value of the residual pressure in the apparatus.

The very problem of designing vacuum systems for technological installations of enterprises of the chemical and petrochemical complex lies at the "junction" of two sciences: vacuum technology and chemical engineering.

Thus, the problem of selection of standard sizes of high-vacuum and forevacuum pumps is urgent, and determination of the system layout is impossible without preliminary calculation of the thermal and material balance of the installation as a whole.

As noted above, the optimal solution to the problem of VGS reconstruction in this production is to replace the existing SEP with a single vacuum generating station based on the LRVP. Since the residual pressure on the suction of the LRVP is limited by the vapor pressure of the service liquid, it is advisable to install a pre-compressed vacuum pump before the LRVP, which would perform pre-compression of the pumped mixture. Mechanical booster pumps are well suited for this purpose [18–20,28].

The plant—which was the subject of this study, has a positive experience of operating VGS based on the LRVP. In the design specifications, there was a point according to which it was necessary to work out three different VGS schemes. Therefore, it was decided to consider the following schemes: the existing one (SEP), and two different LRVP schemes.

The minimum residual pressure at the suction in the LRVP is determined by the saturated vapor pressure of the service fluid. Even if the distillate product of the column is used, in that case, during the compression process in the pump condensation of the condensed components (mainly water vapor) of the pumped-out mixture will occur, which will significantly change the properties of the service fluid.

The mixture leaving the top of the vacuum columns must be compressed to such pressure, at which the LRVP operation is possible. In addition, the volumetric flow rate of the mixture must be within the limits of the operability of a particular liquid–water pump. This can be achieved as follows:

- successively compress the pumped-out mixture in several high-vacuum pumps;
- compress the mixture to a certain intermediate pressure, cool to a temperature at which some of the substances in the mixture begin to condense.

Based on all of the above, when designing the VGS, the schemes that will be described below were considered.

In Figure 3a,b shown two different technological schemes of the reconstructed VGS based on LRVP and mechanical booster pumps.

The uncondensed gases are collected in a single collector and fed to the suction of a mechanical booster pump P-1 (Figure 3a), where the intermediate pressure is compressed. Further, the compressed gases are compressed in a mechanical booster P-2 and fed to the suction in LRVP P-3, where the distillate of the vacuum column is used as a service liquid, which is recirculated through the heat exchanger H-1, into the pipe space of which the refrigerant is also supplied. It is also possible to replace

the pump P-2 with the condenser K-1 (Scheme 2, Figure 3b), in which the mixture compressed in P-1 is cooled and partially condensed, thereby "unloading" the liquid–water pump. The pumped gas and service liquid are separated in the separator S-1.

The calculation schemes of the VGS are shown in Figures 4 and 5. COMP Module (P-1 and P-2 modules in Figures 4 and 5) modeled a booster pump, the compression ratio of which was determined by the characteristic and fixed in the module specification. All processes occurring in the LRVP were calculated using the provisions described in [29].

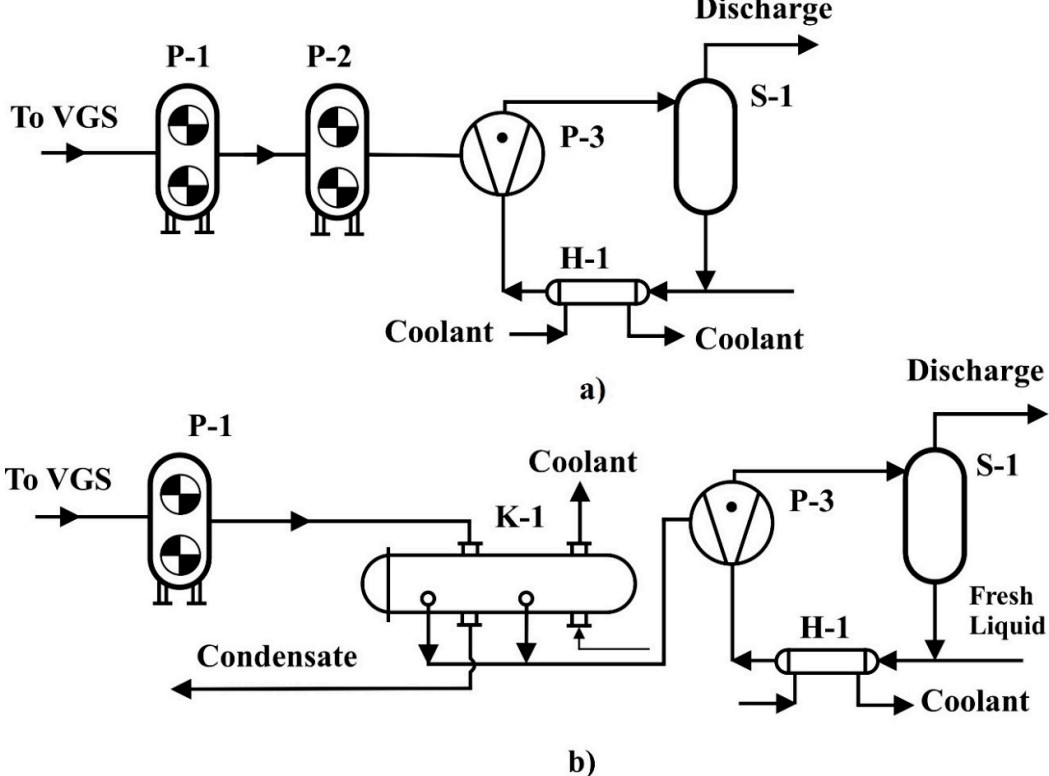

**Figure 3.** Technological Scheme 1 (**a**) and Scheme 2 (**b**) of VGS.

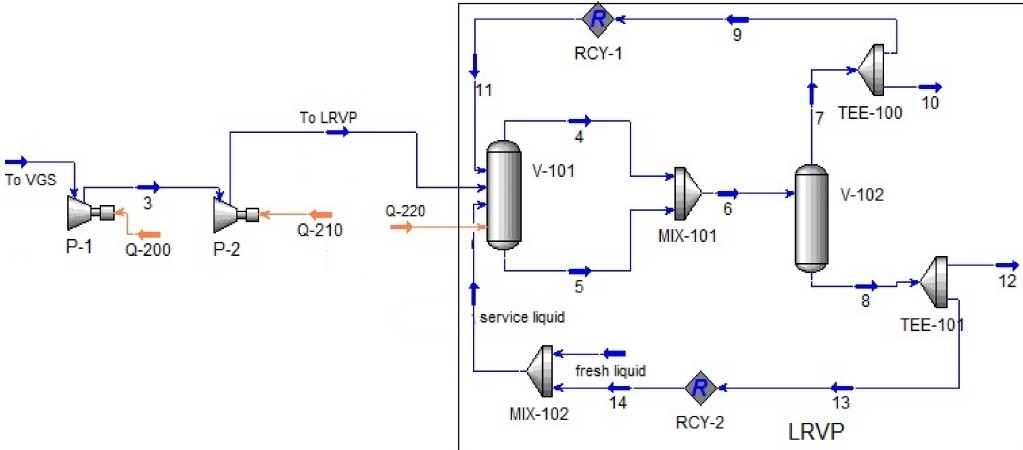

**Figure 4.** Calculation scheme of the VGS synthesized in the Unisim Design R461 software package (Scheme 1). Material flows: 3–to P-2, 4, 6, 7, 9, 11–recirculaled gas; 10, 12–discharge; 5, 8, 13, 14–recirculaled liquid. Heat flows: Q-200, Q-210 and Q-220-drive power P-1, P-2 and P-3, respectively. Modules: V-101, V-102–separator; MIX-101, MIX-102–mixer; TEE-100, TEE-101–divider; RCY-1, RCY-2 recycle module.

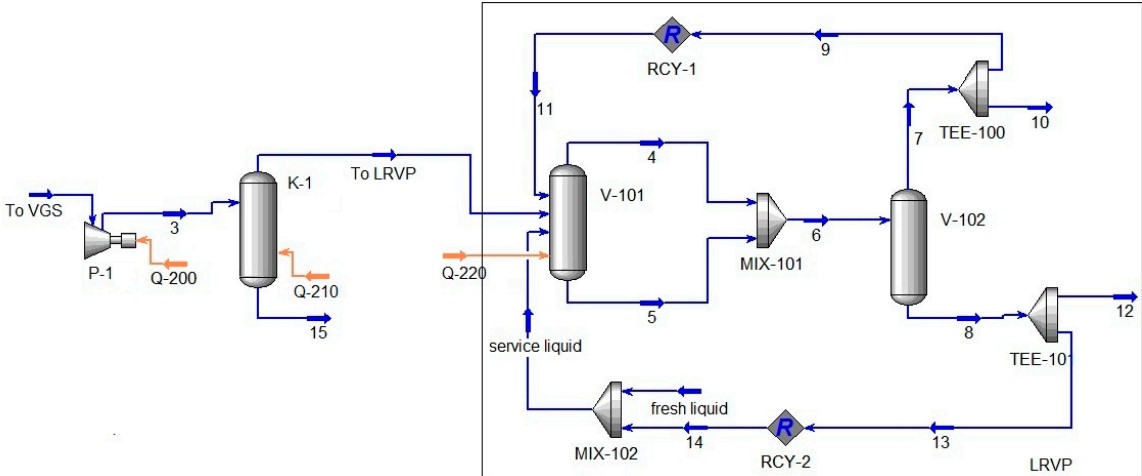

**Figure 5.** Calculation scheme of the VGS synthesized in the Unisim Design R461 software package (Scheme 2). Material flows: 3–to K-1; 4, 6, 7, 9, 11–recirculaled gas; 10, 12–discharge; 5, 8, 13, 14–recirculaled liquid. Heat flows: Q-200, Q-210 and Q-220-drive power P-1, P-2 and P-3, respectively. Modules: K-1, V-101, V-102–separator; MIX-101, MIX-102–mixer; TEE-100, TEE-101–divider; RCY-1, RCY-2 recycle module.

## 4. Identification of Calculation Schemes

A comparison of the calculation results with the production data for the column is presented in Tables 6–8. The calculation data correlates well with the technological parameters of the column, so we can conclude that the models are adequate, and the calculation data is correct in the range under study.

In this case, non-condensable gases enter the vacuum system from three main sources:

- leakage gases entering the system from the environment (usually from the atmosphere);
- gases, contained in the raw material of the plant, in dissolved and condensed form (most often air and water vapor).

Flow gases enter the vacuum system through equipment leaks (welds, flange gaskets, pump seals, etc.). Calculation of these gases is usually based on the condition that during the testing of the equipment for tightness at the operating pressure $P$, the pressure increase over a certain period ($\tau$) should not exceed a certain normalized value ($A_{norm}$):

$$\frac{P_{en} - P_{in}}{P_{in} \cdot \tau} = A_{norm} \tag{14}$$

Obviously, the flow rate of gases will be determined mainly by the volume of equipment to be evacuated. Calculating the flow rate of decomposition gases is a rather complex task [30,31], so their flow rate is usually assigned based on experimental data.

According to the methods [30], the flow rate of flow gases is determined from the equation:

$$W = C \cdot V^B \tag{15}$$

The empirical coefficients of equation $C$ and $B$ (15) were defined for certain equipment and depend on pressure, so their application in this case may also lead to an error. Therefore, the amount of flowing air was determined by conducting a numerical experiment, in which the condensation temperature in the "tail" condensers changed and the output of non-condensing gases was fixed.

Since the suction pressure in the VGS was not precisely determined, and the required pressure at the top of the vacuum columns should not exceed 6.65 mbar (lies on the range 10–15 mbar), required pressure was fixed at the top of the column, and the condensation pressure, based on production

data, was assumed to be 4 mbar. Analysis of the characteristics of steam-ejector pumps according to the GIPRONEFTEASH catalog [32] shows that the amount of the mixture entering the suction into each SEP is 11.5 kg/h of the mixture at a pressure of 4 mbar. Therefore, the flow rate of flow gases introduced into the column by a separate flow was determined based on the condition that the flow rate of non-condensed gases was equal to 11.5 kg/h. The results of the numerical study are shown in Figure 6a–c.

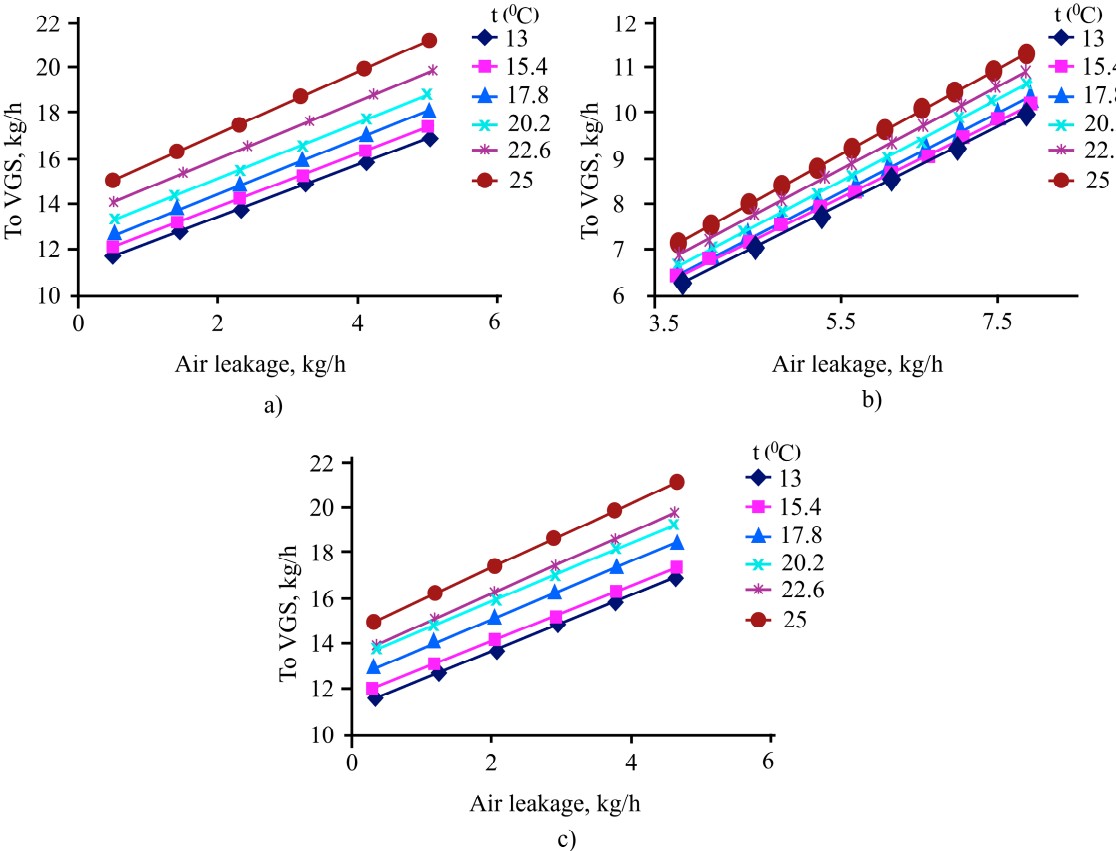

**Figure 6.** Dependence of the yield of non-condensed gases from incoming air and K-40 (**a**), K-56 (**b**) and K-92 (**c**) at different temperatures.

Thus, the K-40 column receives 0.6 kg/h of atmospheric air, K-56—7.8 kg/h, and the K-92 column —0.8 kg/h. Since the K-40 and K-92 columns are similar in their hardware and technological design, the flow rate of the flowing gases was almost the same. The K-56 column has an additional side selection, so the amount of incoming air is greater.

## 5. Determining the Layout of the Reconstructed VGS

Thus, the total load on the VGS will be 9600 m$^3$/h of the mixture, which will mainly consist of air and water vapor. Despite the fact that the vacuum column distillate is supposed to be used as the service liquid of the LRVP, the pump will not develop a residual suction pressure below 30 mbar, so the booster pump must be selected so that the compression ratio of the mixture is at least 8–12.

A feature of the design of technological objects is the need to combine the properties of equipment, which are different in design and principle of operation. Ergo, the choice of general equipment parameters and the search for correlation points in which these devices will work together. For this, equipment parameters can be replaced by "characteristics", which show the general properties of the equipment, and to determine the parameters of correlation, they can be represented in the form of

graphical dependencies. It is obvious that the points of intersection (points of correlation) of these characteristics will be the design parameters of the equipment being developed.

This problem is solved with the best accuracy using computer simulation tools [7,12–17].

The main parameters of vacuum pumps are the dependence of the performance on the suction pressure; therefore, to match the characteristics of the rectification unit and the VGS, it is necessary to build the dependence of the volumetric flow rate of the mixture leaving the tail condensers on pressure, as well as recalculate the passport characteristics of vacuum pumps for new operating conditions.

The calculation and selection of mechanical boosters was carried out according to the following methodology [28]: at several values of the outlet pressure, the inlet pressure and the productivity of the adopted pumps were determined using the dependencies presented below:

$$p_a = \frac{p_v \cdot Q_v}{Q_{eff}} \tag{16}$$

$$Q_{eff} = \eta \cdot Q_{th} \tag{17}$$

$$\eta = \frac{k_0}{k_0 + k_{th}} \tag{18}$$

$$k_{th} = \frac{Q_{th}}{Q_v} \tag{19}$$

The correlation of the characteristics of the equipment for the circuit figure is shown in Figure 7. Decimal logarithms of volumetric flow and pressure values are shown on the graph for easier display.

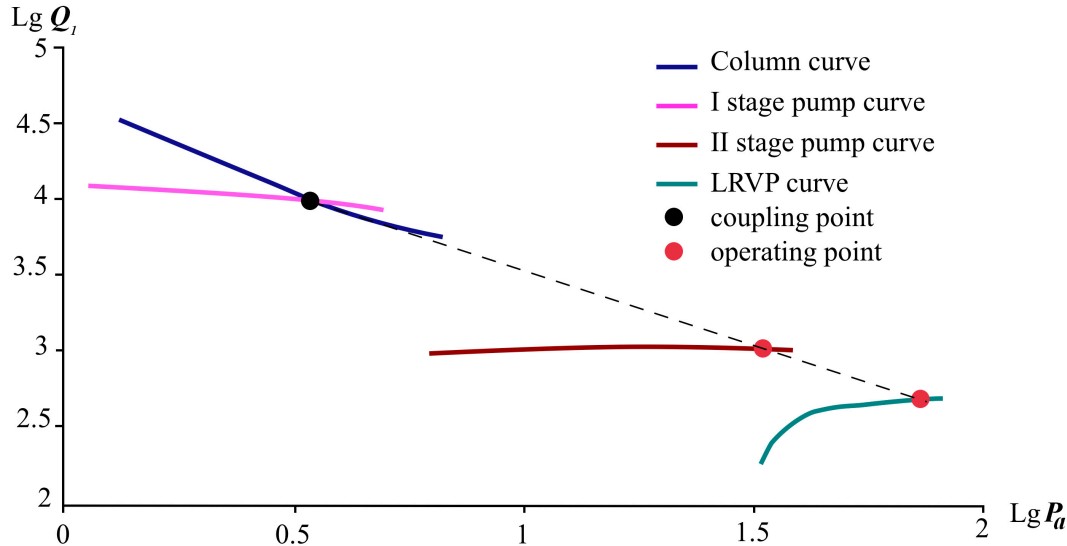

**Figure 7.** Coupled curves of mechanical booster pumps and LRVP (Scheme 1).

According to [33,34], the theoretical pump capacity of the first stage was taken to equal 18,000 m³/h, the second stage—1200 m³/h. The performance of the third stage was determined according to the model shown in Figure 4 in accordance with the provisions set out in [34].

Maximum compression $k_0$ of the Roots pumps Aerzen GMb 17.15HV and Hanbell PR1300 as a function of fore vacuum pressure was selected according with data, presented in [33]. Since the mixture contains a sufficient amount of condensable vapors in its composition, it was decided to install a LRVP as the final stage of compression, and an aqueous solution of monoethanolamine obtained at the installation as a service liquid. Therefore, per the provisions outlined in [19], the characteristics of LRVP were recalculated for the new operating conditions by [29]. Per the provisions outlined in [19], the characteristics of LRVP were recalculated for the new operating conditions by [29].

The results of calculating the stages of the vacuum system are presented in Tables 9 and 10 and on the graph in Figure 7.

**Table 9.** Calculation data for I and II stages (Scheme 1).

| Forevacuum Pressure $p_v$, mbar | Capacity of Hanbell PR1300, m³/h | $K_{th}$ | $K_0$, [28] | $\eta$ | $Q_{eff}$ | Intake Pressure $p_a$, mbar |
|---|---|---|---|---|---|---|
| 38.667 | 993.103 | 18.125 | 18 | 0.498 | 8968.858 | 4.281 |
| 32.801 | 1003.010 | 17.946 | 20 | 0.527 | 9787.168 | 3.468 |
| 26.000 | 1015.385 | 17.727 | 23 | 0.565 | 10,165.179 | 2.597 |
| 20.441 | 1027.338 | 17.521 | 25 | 0.588 | 10,583.004 | 1.984 |
| 17.500 | 1028.571 | 17.500 | 28 | 0.615 | 11,076.923 | 1.625 |
| 13.439 | 1012.010 | 17.786 | 34 | 0.657 | 11,817.780 | 1.151 |
| 8.883 | 985.075 | 18.273 | 43 | 0.702 | 12,632.047 | 0.693 |
| 6.325 | 939.130 | 19.167 | 41 | 0.681 | 12,265.928 | 0.484 |

**Table 10.** Calculation data for II and III stages (Scheme 1).

| Forevacuum Pressure $p_v$, mbar | Capacity of SIHI LPH 65320, m³/h | $K_{th}$ | $K_0$, [28] | $\eta$ | $Q_{eff}$ | Intake Pressure $p_a$, mbar |
|---|---|---|---|---|---|---|
| 80 | 480 | 2.50 | 12 | 0.828 | 993.103 | 38.667 |
| 70 | 470 | 2.55 | 13 | 0.836 | 1003.010 | 32.801 |
| 60 | 440 | 2.73 | 15 | 0.846 | 1015.385 | 26.000 |
| 50 | 420 | 2.86 | 17 | 0.856 | 1027.338 | 20.441 |
| 45 | 400 | 3.00 | 18 | 0.857 | 1028.571 | 17.500 |
| 40 | 340 | 3.53 | 19 | 0.843 | 1012.010 | 13.439 |
| 35 | 250 | 4.80 | 22 | 0.821 | 985.075 | 8.883 |
| 33 | 180 | 6.67 | 24 | 0.783 | 939.130 | 6.325 |

Based on the results of the calculation, the following distribution of the inlet pressure among the stages was determined: at the inlet to the first stage, the pressure will be 4 mbar, in the second stage—38 mbar, in the third—70 mbar. In Figure 7, the resulting pressures are connected by a dotted line.

The calculated flow parameters for the selected operating points are presented in the Table 11 (see flow designations in Figure 4).

**Table 11.** Calculated heat and material balance (Scheme 1).

| Stream Name | Pressure, mbar | T (°C) | Vapor Fraction | Mass Flow Rate, kg/h | Heat Flow, kJ/h | Composition, Mass Fraction | | |
|---|---|---|---|---|---|---|---|---|
| | | | | | | Water | MEA | Air |
| To VGS | 4 | 13 | 1 | | −231,791.145 | | | |
| 3 | 36.86 | 44 | 1 | 33.5 | −230,483.791 | 0.502 | 0.048 | 0.45 |
| To LRVP | 70 | 44 | 1 | | −226,805.676 | | | |
| 10 | | 18.38 | 0 | 17.30 | −1448.092 | 0.009 | 0.000153 | 0.9908 |
| 12 | | | 0 | 31.20 | −232,116.1703 | | | - |
| Fresh liquid | 1013 | 18 | 0 | 60,000 | −446,397,374.3 | 0.5514 | 0.4486 | |
| Service liquid | | | 0 | 15 | −160,371.99 | 0.55 | 0.45 | - |

The characteristics of the selected equipment are presented in the Table 12 [32,33].

**Table 12.** Technical characteristics of the selected equipment (Scheme 1).

| Pump Brand | Resource Consumption |
|---|---|
| **First Stage** | |
| Double rotor pump Aerzen GMb17.15HV | Productivity: 15,685 m$^3$/h; Drive power—30 kW; Suction working pressure—4 mbar. |
| **Second Stage** | |
| Double rotor pump Hanbell PR1300 | Productivity: 1050 m$^3$/h; Drive power—4 kW; Suction working pressure—38 mbar. |
| **Third Stage** | |
| Liquid ring vacuum pump SIHI LPH 65320 | Performance: Dry air—370 m$^3$/h; By air saturated with water vapor—420 m$^3$/h; Working conditions—470 m$^3$/h. Drive power—8 kW; The number of revolutions—1450 rpm; Suction working pressure—70 mbar. Service liquid consumption—2.7 m$^3$/h. |

The interface conditions between the pre-connected booster pump and the LRVP (Scheme 2) will determine the temperature and condensation pressure of K-1 vacuum condenser. Therefore, to determine the size of the condenser and the conditions for condensation of the mixture, a numerical experiment was performed, in which the condensation temperature in the V-103 module changed. The results are shown in Figure 8.

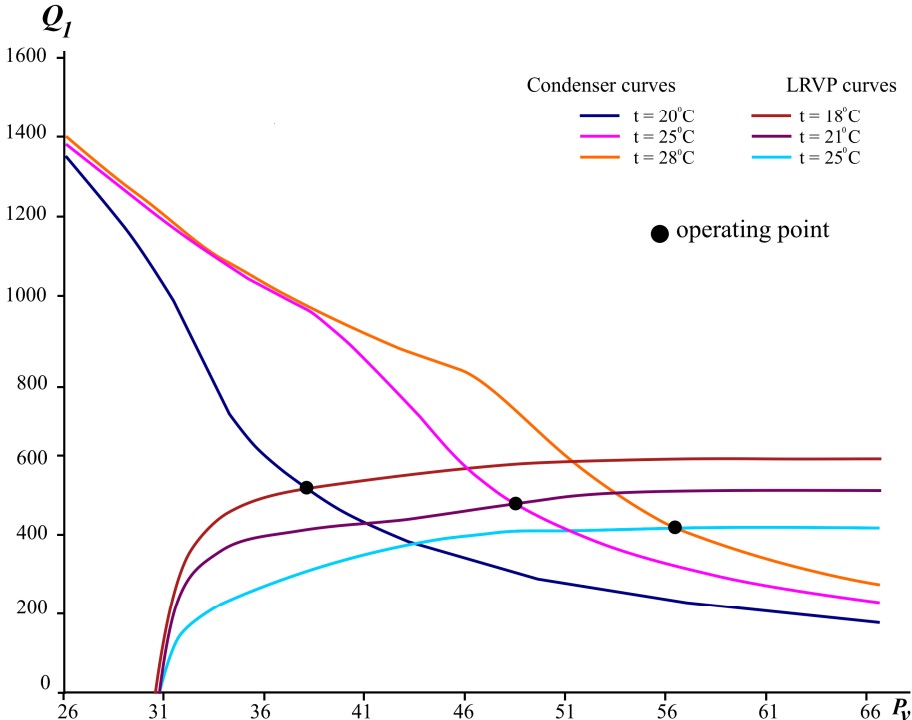

**Figure 8.** Output non-condensed gases from the condenser K-1 and features LRVP depending on pressure.

As can be seen from the graph, the curves of the characteristics of the condenser and LRVP intersect at some points (coupling points, see Figure 8). This means that at these points the volumetric flow rate of the medium leaving the condenser and the working capacity of the LRVP become equal. Such points can be taken as design points for operating residual pressure. If we take into account that the first stage compresses the mixture to 38 mbar, then it can be seen from the graph that at a condensation temperature of 20 °C and an assumed temperature of the service liquid of 18 °C, the proposed LRVP SIHI LPH 65,327 will be efficient and will provide the required outlet pressure of P-1.

The calculated flow parameters for the selected operating points are presented in the Table 13 (see flow designations in Figure 5).

**Table 13.** Calculated heat and material balance (Scheme 2).

| Stream Name | Pressure, mbar | T (°C) | Vapor Fraction | Mass Flow Rate, kg/h | Heat Flow, kJ/h | Composition, Mass Fraction | | |
|---|---|---|---|---|---|---|---|---|
| | | | | | | Water | MEA | Air |
| To VGS | 4 | 13 | 1 | 33.5 | −231,791.145 | 0.502 | 0.048 | 0.45 |
| 3 | 36.86 | 44 | 1 | 33.5 | −230,483.791 | | | |
| To LRVP | 36.86 | 20 | 1 | 21.26 | −77,063.476 | 0.2688 | 0.0002 | 0.731 |
| 15 | 36.86 | 20 | 0 | 13.24 | −191,537.329 | 0.8782 | 0.1218 | - |
| 10 | 1013 | 18.25 | 0 | 17.75 | −1476.516 | 0.0056 | 0.0003 | 0.9941 |
| 12 | | | 0 | 18.51 | −137,704.197 | 0.5511 | 0.4488 | - |
| Fresh liquid | | 18 | 0 | 60,000 | −446,397,374.3 | | | |
| Service liquid | | | 0 | 15 | −160,371.995 | 0.55 | 0.45 | - |

Parameters of the selected hardware are shown on Table 14 [33,34].

**Table 14.** Parameters of the selected hardware (Scheme 2).

| Equipment Brand | Parameter |
|---|---|
| **First stage** | |
| Aerzen gmb17.15HV twin-rotor pump | Capacity: 15,685 m³/h; Drive power—30 kW; The working pressure at the suction is 3.99 mbar. |
| **Condenser** | |
| Vacuum shell-and-tube condenser 600 BEM-2-M1/25G-3-2-U-I | Diameter—600 mm; Tube passes—2; Tube—3 m. |
| **Second Stage** | |
| SIHI LPH 65,327 liquid ring vacuum pump | Suction pressure—38 mbar. Performance: In the dry air of 500 m³/h; By air saturated with water vapor—700 m³/h; According to operating conditions—525 m³/h; Rotary speed—1740 rpm; Drive power—10 kW. |

## 6. Feasibility Study

To achieve optimal technical and economic indicators, the standard sizes of equipment were determined by point 2. The standard sizes of the selected equipment are shown in Tables 5 and 6.
Total for columns K-40, K-56, and K-90:

- 0.6 m$^3$/h of chemically polluted wastewater is generated;
- 516 kg/h (0.35 Gcal/h) of medium-pressure water vapor is consumed;
- recycled water—30 m$^3$/h.

The positive economic effect is planned to be achieved by eliminating the use of water vapor, reducing the consumption of recycled water, and eliminating the formation of chemically polluted runoff. The total energy consumption will be:

- for Scheme 1: 42 kW of electricity and 2.7 m$^3$/h of coolant water;
- for Scheme 2: 40 kW of electricity and 3 m$^3$/h of coolant water.
- Wastewater is not formed in both schemes.

Analyzing the table, we can conclude that the payback period of the proposed solutions is less than five years, which makes it economically attractive for the customer. System 1 is technically more complex; however, it is more attractive, since to install a vacuum condenser it is necessary to ensure its rise to a level of at least 10 m above the ground, which complicates the project.

The calculation of operating costs was carried out according to the following method: the formula was used to determine the costs of consumed resources:

$$Cost_{RE} = \Pr_{RE} \cdot TC_{RE} \cdot h \tag{20}$$

Furthermore, all costs were summed up and the total operating costs were determined:

$$Cost_{TRE} = \sum_{RE} Cost_{RE} \tag{21}$$

After that, the economic effect of the implementation is determined, which is calculated by subtracting from the current operating costs for the operation of the proposed option:

$$EE = \sum_{RE} Cost_{RE_{CS}} - \sum_{RE} Cost_{RE} \tag{22}$$

After that, with the accepted capital costs, the payback period was determined

$$PP = \frac{CE}{EE} \tag{23}$$

The calculation results are summarized in the Table 15.

**Table 15.** Economical comparison.

| Parameter | Units | Current System | Scheme 1 | Scheme 2 |
|---|---|---|---|---|
| Operational hours per year | hour | 8000 | 8000 | 8000 |
| Total steam consumption | Gcal/h | 0.342 | 0 | 0 |
| Recycled water | m³/h | 30 | 2.7 | 3.00 |
| Wastewater | m³/h | 0.6 | 0 | 0 |
| Electricity | kW | 0 | 42 | 40 |
| **Expenses** | | | | |
| Steam | $/year | 49,060.44 | 0 | 0 |
| Recycled water | $/year | 5618.10 | 505.63 | 561.81 |
| Wastewater | $/year | 1063.65 | 0.00 | 0.00 |
| Electricity | $/year | 0.00 | 12,814.31 | 12,204.10 |
| Expenses | $/year | 55,742.19 | 13,319.94 | 12,765.91 |
| Economic effect | $/year | 0 | 42,422.25 | 42,976.28 |
| Capital Expenses | $ | 0 | 175,500.00 | 154,322.00 |
| Payback period | year | 0.00 | 4.14 | 3.59 |

## 7. Conclusions

As part of the task (determining the layout of the vacuum system for distillation columns for separating a mixture of ethanolamines and calculating the payback period of capital costs), it was proposed to use the Unisim Design R461 modeling package to calculate the heat and material balances, which are well suited for work of this kind. Despite the fact that this software package does not have a module for calculating a vacuum system, this task can be solved by methods of system analysis using a combination of standard modules included in the program database.

A numerical experiment on the developed model of vacuum columns investigated the influence of the flow rate of incoming gases (atmospheric air) on the flow rate of gases leaving the vacuum system and it was found that the flow rate of the flowing gases into the K-40 column will be 0.6 kg/h, in the K-56 column—7.8 kg/h, and in K-92 column—0.8 kg/h. The total load on the single vacuum station will be 9600 m³/h.

When determining the layout of the main elements of the system, the following task arises: to link into a single whole, various types of main equipment, which are different in nature. To do this, it was necessary to determine those characteristics that are common to the machines and devices under consideration. As such characteristics, it was proposed to take the dependence of the flow rate on pressure, and for two-rotor pumps (Scheme 1), both input and output flow characteristics were used, and for a vacuum condenser, the dependence of flow characteristics on both pressure and temperature. The points at which these characteristics intersected were taken as design points, along which, the main technological equipment was selected.

After determining the layout of the system, the consumption of consumed energy resources and the total economic effect of the proposed design solutions were calculated. It turned out that for Scheme 1 the economic effect will be $42,422.25/year, and for Scheme 2—$42,976.28/year. Thus, if we take Scheme 1 as a basic modernization project, then the project will pay off in 4.14 years, and if Scheme 2—in 3.59 years.

Since the pumped medium contains a significant amount of condensable components (ethanolamines and water), special attention should be paid to the condensation conditions, which are determined by temperature and pressure, when designing installations of this kind. However, when choosing

these characteristics, uncertainty arises in the choice of these parameters, which in turn depend on the peculiarities of the interaction of the main blocks of the system. So, the article considered types of VGS, which differ from each other in that reducing the load on the forevacuum pump is achieved in two different ways—by increasing the pressure (Scheme 1) and lowering the temperature (Scheme 2). It turned out that the decrease in the load gave similar results: in the first case, the load dropped to 470 $m^3$/h, and in the second, to 525 $m^3$/h. At the same time, although the size of the forevacuum pump increased, the complexity of the system decreased, which led to a decrease in operating costs.

The maximum volumetric flow rate of the mixture enters the first stage pump, which makes this equipment the largest and most energy-consuming. If we consider measures to reduce the load on the first stage, then an increase in the pressure in the columns is impossible (the decomposition of the initial mixture will begin in the bottoms), and a decrease in temperature is possible only when using cooling systems with artificially generated cold. In this case, it becomes possible to control the operation of both the vacuum system and the vacuum columns.

Based on all of the above, we can conclude that to achieve better performance for various types of vacuum systems, lowering the temperature of the inlet flows is a more effective technique than increasing the pressure.

The methods proposed in the article for solving the problem can be applied in the design of new and reconstruction of existing vacuum systems for technological installations.

**Author Contributions:** Conceptualization, E.O., methodology, E.O.; validation, E.T. and E.O.; formal analysis, E.T. and S.P.; data curation, S.P.; writing—original draft preparation, E.O.; writing—review and editing, E.T.; visualization, E.O.; supervision, E.T.; funding acquisition, S.P. All authors have read and agreed to the published version of the manuscript.

**Funding:** This research was funded by the Ministry of Science and Higher Education of the Russian Federation grant number 075-00315-20-01 «Energy saving processes of liquid mixtures separation for the recovery of industrial solvents»

**Acknowledgments:** We thank Daniel Bugembe for assistance with the translation of this paper and research.

**Conflicts of Interest:** The authors declare no conflict of interest.

## Nomenclature

| | |
|---|---|
| K-40, K-56, and K-90 | vacuum distillation columns; |
| $A_{norm}$ | normalized value; |
| $W$ | air leakage, kg/h; |
| C, B | empirical coefficients that depend on pressure; |
| $V$ | volume, $m^3$; |
| $P$ | pressure, mbar; |
| t | temperature, °C; |
| $Q_1$ | volume flow rate, $m^3$/h; |
| q | pure component area parameter; |
| $Q$ | group area parameter; |
| r | pure component volume parameter; |
| $R$ | gas constant (without subscript); |
| $R_k$ | group volume parameter (with subscript); |
| T | temperature, K; |
| $u_{ji}$ | UNIQUAC binary interaction parameter; |
| $U_{nm}$ | UNIFAC binary interaction parameter; |
| $x$ | liquid phase mole fraction; |
| X | liquid phase group fraction; |

| | |
|---|---|
| $y$ | vapor phase mole fraction; |
| z | lattice coordination number, a constant here set equal to ten; |
| $P_a$ | intake pressure; |
| $P_v$ | forevaccuum pressure; |
| $k_o$ | coefficient that depends on $P_v$; |
| $Q_{eff}$ | effective capacity, $m^3/h$; |
| $Q_{th}$ | theoretical capacity, $m^3/h$; |
| $\eta$ | volumetric efficiency; |
| $k_{th}$ | grading coefficient; |
| **(o)** | operational data; |
| **(s)** | simulation data; |
| Pr | price, $; |
| TC | total consumption; |
| h | operational hours per year, hour; |
| *EE* | economic effect; |
| CE | capital expenses; |
| *PP* | payback period. |

**Greek letters**

| | |
|---|---|
| $\gamma_i$ | activity coefficient of component *I*; |
| $\Gamma_k$ | activity coefficient of group *k*; |
| $\Gamma_k^{(i)}$ | activity coefficient of group *k* in pure component *i*; |
| $\Phi_i$ | segment fraction of component *i*; |
| $\upsilon_k^{(i)}$ | number of groups of kind *k* in a molecule of component *i*; |
| $\theta_i$ | area fraction of component *i*; |
| $\Theta_k$ | area fraction of component *k*; |

**Subscripts**

| | |
|---|---|
| sl | service liquid; |
| *i, i, k* | component *i, j,* and k; |
| *k*, m, n | group k, m, and n; |
| RE | resource; |
| CS | current system; |
| T | total. |

**Superscripts**

| | |
|---|---|
| C | combinatorial; |
| R | residual. |

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
