# Peer review of "Coupled Simulation of a Vacuum Creation System and a Rectification Column Block"

_processes, doi:10.3390/pr8111333_

Round 1

Reviewer 1 Report

This problem is relevant for journal scope. The concept and aim are clearly defined. The manuscript is well written, I could not find typing errors. The manuscript follows the formal regulations of journal.

I suggest the acceptance after major revision.

Remarks, suggestions, questions

  1. Please cite more papers from this journal at the last two years in the similar topic of this research.
  2. Please add Nomenclature part to the manuscript.
  3. The abstract and conclusion parts are too short and general. Please expand it with specific data and explanations.
  4. Please add some information about efficiency about vacuum creation system and rectification columns.
  5. What does it mean the following abbreviations: K-40, K-56, and K-92? (40, 56 and 92 numbers)
  6. Please give VLE data of investigated mixtures to the manuscript (diagram and thermodynamic coefficients).
  7. Please add the references of Table 1 and Table 6.
  8. How did you select the location of feed plate?

Reviewer 2 Report

  1. Title of Table 4: Technological (not Ttechnological)
  2. Table 4: the symbol "Reflux*" is used twice
  3. Table 2-4: It would be useful to describe exactly what the streams marked with * are about .Probably, as the text shows, they apply proposed b) blocks from Figure 1?
  4. "After distilling water and residual ammonia in the K-20 apparatus, the dehydrated flow of ethanolamines is fed between the second and third sections of the K-40 column": In the Figure 2 feed to K-40 is descrine as "OT k-29".   By the way, why "OT" instead "From"?
  5. Equation 2: No description of variables in equation 2 - you can of course guess, but it was enough to use symbols V,W,C in the text above and below the equation.
  6. Figure 6 caption: "K40), K-56 b) and K-92)" instead K-40 a), K-56 bb) and K-92 c).
  7. Title of Table 6: "Parameters of the selected hardware (scheme 2)", while parameters in the Table 6 are given for both scheme 1 and scheme 2.

Reviewer 3 Report

The authors present a study on vacuum creation systems for chemical processes. Anyhow this is an important topic with respect to energy-efficiency, the manuscript falls short, to discuss this question from a methodological point of view. Alltogether the manuscript rather appears to be a technical report than a scientific study. Additionally, formatting and writing is not sufficient.

Some more detailed aspects:

Introduction: Neither the specific research question nor sufficient relation to other scientific work in this field is presented. The authors do not give a clear statement on what to expect in this mansucript.

Methods used are not presented in adaquate manner.

All Figures: Low resolution, quality need to be improved

Page 2, Line 62: This conclusion is not derived straight forward.

Table 1 and Table 6: . instead of ,

Page 3, Line 1: Which program was used exaclty? In the way described by the authors it is not clear at this point, anyways the authors name the used tool later.

Page 4, Line 129: On what methodological bases do the different schemes for the overall system originate? The argumentation is not comprehensible.

Tables 2-4: What is indicated by unsing *? Also this tables are formatted in a unclear way. Where do the ranges in compositions, temperatures and mass rates result from?

Desing parameters of the columns are missing (reflux ratios, feedstages, duties,...)

Abbreviations are used without further explaination.

What kind of thermodynamic model is used for the simulation?

Page 6, Line 165: It is not clear what are the technological parameters of the columns and how this statement can be derived from the values given in Table 2-4.

Page 7, Line 218: What exaclty is meant by this statement?

Figure 7: Units of Q and P are missing. Coupling point is not explained in the text. What is meant here?

A detailed discussion of the results is missing.

The conclusion falls short to put the overall research conducted into a bigger picture and to critically discuss further questions, e.g.: How can the results obtained be transferred to other processes operated under vacuum conditions?

Throughout the manuscript pressures should be given in bar or Pa.

Reviewer 4 Report

The main focus of this work is to deal with the coupled simulation of a vacuum creation system and a rectification column unit for separating a mixture of ethanolamines to reduce the operating costs of the vacuuming process. However, its hard to find real contribution of this work. There is no new design and optimization methodologies in this paper. The results, discussion and conclusions are poor. Furthermore, the paper was not prepared carefully with many typing and grammar errors. Thus, this paper cannot be recommended to be published as it does not meet the minimum quality requirements of Processes journal.

Author Response

We regret that the innitial version of the manuscript that we submitted had several errors and areas in which it was found to be lacking these areas included but not limited to the results, discussion and conclusions sections of the research work. We revisted the whole manuscript and greatly improved its content and the way it is presented in order to meet the  quality requirements of Processes journal.

Round 2

Reviewer 1 Report

I have studied the manuscript and the answers. In my opinion the answers are professionally well-founded. I suggest the acceptance in this present form for publication.

Author Response

Thank you for your feedback on our article, your recommendations helped us significantly improve our manuscript.

Reviewer 3 Report

Although the authors did improve the readability of their manuscript there are still some drawbacks that need to be addressed:

  • The introduction still falls short to frame the research question to be answered in a precise and condensed form.
  • An overview on the structure and what to expect is still missing.
  • The quality of Fig. 7 and 8 is still not sufficient
  • Formatting (especially of units e.g. m3 -> m³) must be corrected.
  • A consistency in using , or . as decimal separator is still missing. Some tables use , and others . This needs to be harmonized
  • A careful spell check is needed as there are still some errors.

Reviewer 4 Report

Some general and specific points are necessary to provide a better understanding of the work for the reader as well as to get a better idea of the actual contribution of this work:

  1. The authors should clearly establish which the novelty of the work is.
  2. The authors should show information of energy and main streams, such as flowrate, composition, temperature, pressure in Figures 4, 5, and others.
  3. The author should emphasize what advantages and disadvantages of proposed solution.
  4. The author should show detail calculation of investment and operating costs in Supplementary material.
